# On Whether Ca-125 Is the Answer for Diagnosing Overhydration, Particularly in End-Stage Kidney Disease Patients—A Systematic Review

**DOI:** 10.3390/ijms25042192

**Published:** 2024-02-12

**Authors:** Barbara Emilia Nikitiuk, Alicja Rydzewska-Rosołowska, Katarzyna Kakareko, Irena Głowińska, Tomasz Hryszko

**Affiliations:** 2nd Department of Nephrology, Hypertension, and Internal Medicine with Dialysis Unit, Medical University of Bialystok, 15-276 Bialystok, Poland; bn120297@gmail.com (B.E.N.); katarzyna.kakareko@umb.edu.pl (K.K.); irena.glowinska@umb.edu.pl (I.G.); tomasz.hryszko@umb.edu.pl (T.H.)

**Keywords:** fluid status, overhydration, hydration status dialysis, renal failure, kidney failure, heart failure, Ca-125, Ca-125 kidney disease, VEXUS, BIS

## Abstract

Overhydration (OH) is a prevalent medical problem that occurs in patients with kidney failure, but a specific marker has still not been found. Patients requiring kidney replacement therapy suffer from a water imbalance, which is correlated with mortality rates in this population. Currently, clinicians employ techniques such as bioimpedance spectroscopy (BIS) and ultrasound (USG) markers of overhydration or markers of heart and kidney function, namely NT-pro-BNP, GFR, or creatinine levels. New serum markers, including but not limited to Ca-125, galectin-3 (Gal-3), adrenomedullin (AMD), and urocortin-2 (UCN-2), are presently under research and have displayed promising results. Ca-125, which is a protein mainly used in ovarian cancer diagnoses, holds great potential to become an OH marker. It is currently being investigated by cardiologists as it corresponds to the volume status in heart failure (HF) and ventricular hypertrophy, which are also associated with OH. The need to ascertain a more precise marker of overhydration is urgent mainly because physical examinations are exceptionally inaccurate. The signs and symptoms of overhydration, such as edema or a gradual increase in body mass, are not always present, notably in patients with chronic kidney disease. Metabolic disruptions and cachexia can give a false picture of the hydration status. This review paper summarizes the existing knowledge on the assessment of a patient’s hydration status, focusing specifically on kidney diseases and the role of Ca-125.

## 1. Introduction

Ca-125 is a high-mass glycoprotein that is produced by ovarian cancer cells and is used to detect and monitor this disease. Elevated levels of this marker are also seen in patients with lung, uterine, breast, and gastrointestinal tract cancer [1]. Ca-125 originates from the coelomic epithelium and reacts with various structures, especially those undergoing an inflammatory process [2]. It accumulates on the luminal cell side, which releases Ca-125 into the bloodstream when it is disrupted by an invasive cancer. Elevated Ca-125 levels can also be found in non-malignant diseases such as digestive tract cancer, non-Hodgkin lymphoma, acute leukemia, endometriosis, ascites, pelvic inflammatory diseases, tuberculous peritonitis, and pericarditis [1]. It is also being researched by cardiologists as its levels increase in patients with heart failure. As an OH marker, it has also been found to increase in patients with kidney disease. Human peritoneal mesothelial cells are also capable of producing Ca-125, and it has been studied in peritoneal dialysis fluid. The Ca-125 levels in this fluid can be used in clinical situations to predict the extent of peritoneal membrane damage [3]. Ca-125 can be produced by mesothelial cells that have detached from the peritoneal membrane. The effect of dialysis catheter implantation, intra-abdominal surgery, and peritonitis on Ca-125 levels should also be investigated [4]. The clinical situations that are associated with an increase in this marker and their possible causes are listed below in Table 1.

As mentioned above, Ca-125 can increase due to various cancerous and non-cancerous states that stimulate inflammation. OH also triggers the immune system and can cause an increase in inflammatory markers, such as CRP or IL-6, which can be partially responsible for the increase in Ca-125 levels. Hydration status estimation is extremely important for patients with kidney and heart failure because the excess water indicates a deterioration of organ function and correlates directly with patient mortality. Chronic kidney disease (CKD) is a progressive disease that affects more than 10% of the worldwide population and is on the rise [8]. It is well known that metabolic alterations, like hyperglycemia, chronic inflammatory processes, nephrotoxin accumulation, lipid metabolism disruption, or OH, increase the progression of kidney disease. Total body water (TBW) constitutes approximately 50% to 60% of an adult’s total body weight, and it represents over 73% of their lean body mass [9]. A person’s hydration status varies according to their muscle mass, fat mass, electrolyte balance, or physiological states like pregnancy. Naturally, the percentage of TBW tends to decrease with age due to decreased muscle mass and metabolism. Overhydration has been recognized as a known risk factor for hypertension, HF, and increased mortality in patients with kidney failure [9,10,11,12]. OH has been recognized as an independent risk factor for CKD progression in patients with type 2 diabetes mellitus (T2DM) [8]. A study by Zoccali et al. demonstrated that overhydrated patients with end-stage kidney disease (ESKD) have a 62% higher risk of mortality compared to non-overhydrated ones [10]. Similarly, a study by Hung et al. revealed that among a research group of 338 patients with CKD, only 48% were euvolemic [11]. Zoccali, Hung et al. were not the only researchers to acknowledge and describe the dependency between normohydration and long-term survival in patients with kidney diseases [10,11,12,13,14,15,16,17,18,19,20,21,22,23,24,25,26,27,28,29,30]. Patients suffering from overhydration are also more prone to developing peritonitis [31]. Taking this into consideration, it presents a significant challenge for modern physicians to precisely assess and treat OH in CKD patients, as it can lead to irreversible damage to the body and consequently to death [23,27,30]. The clinical symptoms of OH may include distal and proximal swelling, dyspnea, hypertension, and jugular venous pressure. During the physical examination, which includes auscultation, palpation, and percussion, doctors may also hear cracklings in the lungs, particularly in the lower parts. In addition, hepato-jugular reflux might be observed when the pressure applied to the liver causes a sustained rise in the standard blood pressure or jugular blood pressure, which can be observed as an under-skin pulsation. The symptoms may be present not only in OH caused by kidney failure, but also in heart or liver diseases, infections, and carcinogenesis. The presence or absence of clinical symptoms does not exclude OH in patients with kidney diseases. In Eng et al.’s study on a pediatric ESKD population, 25% of the children were hypertensive but not overhydrated [32]. Hypertension has multiple causes, but in ESKD populations, it is mainly exacerbated by OH due to vasodilatation and the stimulation of RAA hormones. This is why blood pressure measurements should be accompanied by further clinical examinations to exclude the eventual influence of OH on the patient’s blood pressure. One of the first symptoms of OH is the presence of pitting pedal edema, but this symptom can also be caused by stasis or excessive vascular permeability [19]. A cross-sectional study of hemodialysis patients established that pitting pedal edema correlates with the body mass index but does not reflect the volume status. Individuals who suffer from ESKD may also develop HF, hepatic failure, and pneumonia, which additionally influences volume status. Longstanding volume overload alerts specific gene expression that leads to the overproduction of endothelial growth factors, changing the vessel’s wall morphology, which leads to increased stiffness [33]. The excess fluid leads to heart fibrosis and left ventricular hypertrophy (LVH), which is the main cause of death due to cardiac malfunction [32,34]. This makes the physical examination extremely difficult to interpret, as the visible symptoms can be due to factors other than the fluid status. Patients’ diagnosis should be confirmed with the use of additional available clinical tests, for example, estimation of the dry body weight, which, according to a definition proposed by Daugirdas et al., includes shortest post-dialysis recovery time, least intradialytic hypotension/symptoms, longest patient survival, fewest cardiovascular/cerebrovascular events, fewest hospitalizations, hypovolemia-related access thromboses, and post-dialysis falls [35]. Doctors should have a holistic approach to the OH problem and combine various available methods to establish the patient’s hydration status and manage between methods depending on the individual’s needs, particularly in the case of cardiorenal syndrome and HF [30]. Present-day medicine encompasses assessing dry body weight through blood pressure, signs and symptoms of OH, blood volume monitoring, and bedside ultrasound [36]. The gold standard for testing body composition and body water distribution in healthy, non-overweight populations is called bioimpedance spectroscopy (BIS). It allows for measuring intracellular and extracellular water, helping to accurately determine cell mass [37]. This method is noninvasive as it only requires the placement of electrodes on patients’ bodies to measure tissue conductivity, hence the water balance [37]. The electrical resistance of external cell water increases during OH as the mass and size of the cells and consequently the length of the pathway around the cells increase. Those advantages have also made BIS a tool applied to dialysis patients. However, Moissl et al. indicated that BIS calculations should be revised to be more precise in an overhydrated, dialyzed population [38]. Among patients who undergo dialysis, results vary according to the researcher and the device used [39]. It has also limited clinical application in certain conditions, not only connected to kidney diseases. This method cannot estimate the accurate water cell ratio in pregnant women with multiple volume changes over time. Its usefulness in patients after limb amputations or with implanted electronic devices like a pacemaker is also diminished. While effective on healthy individuals, this approach exhibits variability in results when applied to clinical populations [30,37]. BIS seems to be a gold standard in theory, but in practice, not all clinics or dialysis stations have the necessary equipment. This sophisticated and modern tool is relatively expensive, rendering it financially unfeasible for some public healthcare institutions.

Nephrologists also use laboratory markers, such as NT-pro-BNP or BNP, to help determine the hydration status. The primary difference between the two markers is that BNP is a biologically active marker, whereas NT-pro-BNP is not. The inactive pro peptide is more stable in the bloodstream, and it correlates strongly with heart failure, cardiovascular congestion, and death [36]. Adrenomedullin and its derivative pro-adrenomedullin (MR-pro-AMD), both of which are markers of endothelial damage, are being actively studied mainly in populations with HF, sepsis, or kidney failure. Both tend to correlate with clinical state and available indirect markers of OH like NT-pro-BNP.

Gal-3 appears to be more responsive to kidney failure and mortality rate than OH itself but shows a precise correlation during the clinical and pre-clinical studies as an organ failure indicator. Ucn-2 is a protein sensitive to vasodilation due to an increase in fluid volume but it gives diverse results when considered as a strict OH marker. Volume status can also be assessed with imaging methods. POCUS (point-of-care ultrasonography) allows the observation of the signs of OH, such as the presence of fluid in the pleura or peritoneum; higher diameter of the jugular vein; IVC (inferior vena cava); abnormal flow through the hepatic vein, portal vein, or renal vein; and estimation of pressure within these structures [39]. A specific protocol called the Venous Access Ultrasound Score (VEXUS) has also been developed and validated.

The holistic approach to the estimation of hydration status in ESKD patients is illustrated in Figure 1.

A precise hydration status marker is still lacking, but it is highly required to improve the available methods of treatment. Scientists should look for an ideal marker, and then doctors should focus on the overall clinical state. Volume overload is harmful to patients, but aggressive fluid removal entails the danger of a decrease in residual renal function and further organ destruction [34].

## 2. Materials and Methods

The main aim of this meta-review is to determine whether modern medicine needs a new overhydration marker. We used the SPIDER method, identifying samples and phenomena of interest as an overhydrated ESKD population and, due to the scarcity of data, including studies of all design and research types rather than evaluating them.

We undertook a systematic search through the PubMed database from the beginning of October 2022 to the end of December 2023. We searched both with individual keywords with all subheadings included. Individual keywords used were “overhydration renal disease”, “overhydration dialysis”, “overhydration bioimpedance”, “Ca-125 renal”, “Ca-125 dialysis”. The results were merged, and duplicates were discarded. The remaining articles were screened for relevance (based on their title, abstract, or full text). Articles were included only if they were clearly related to the subject matter and published in English. There were no date restrictions. Studies assessing OH status in the ESRD population were investigated. Analyzed and selected studies had to include patients with ESKD, on dialysis and non-dialysis treatment. The data from different studies were compared according to follow-up time, study duration, and coherence of the OH estimation. Studies that included patients with acute kidney disease (AKI) were excluded from the analysis due to the complexity of the illness pathogenesis.

## 3. Results

Of the 1760 identified in PubMed records, 1415 were screened, 214 were retrieved and assessed for eligibility, and 108 were finally included in the review. The entire selection process is illustrated in Figure 1. The main features of the original articles on the topic under study are summarized in Table 1 and Table 2.

Potential biases of the chosen studies are the fact that renal diseases can vary depending on race and gender. There are few available studies on child populations; an increase in Ca-125 levels in PD patients due to serosal inflammatory reactions in the abdomen can also be present [41]. Concomitant heart disease also increases the levels of blood markers, particularly NT-pro-BNP. It is not easy for clinicians, as well as scientists, to clearly estimate whether heart disease is a consequence of ESKD or whether are there other complementary factors.

All methods of hydration status estimation are listed below in Table 2.

Currently, as shown in Table 2, the best strategy for volume status assessment is to combine many methods together with the patient’s examination. None of the listed markers is specific to OH status. It is crucial to develop, investigate, and combine all available methods and their correlations. Table 3 lists research papers that combine some of the methods mentioned in this article and their correlation with OH.

As can be noticed in Table 3, most of the studies on OH status estimation are based on BCM, together with the addition of NT-pro-BNP or USG. There are no studies that combine all possible methods of analysis; not many papers include Ca-125 and eventual correction based on BIS or USG. Only two studies included in the paper tested both blood markers.

## 4. Discussion

### 4.1. Ca-125 in Overhydration

Carbohydrate antigen 125 (Ca-125) is a complex glycoprotein that is widely used in cancer diagnosis, especially ovarian cancer [7]. It is mainly synthesized by mesothelial cells in the pericardium, peritoneum, or pleura [7]. It is not known exactly why cells produce Ca-125, but it appears to be stimulated by inflammatory processes and mechanical injury [1,30]. It has recently emerged as a promising marker for congestive HF [7]. As seen for New York Heart Association (NYHA) stage I/II to stage III or IV [30], it increases with a decline in heart function. In the study conducted by Arik et al. considering various cancer markers and their correlation with kidney failure, only Ca-125 and Ca 19.9 were found to be significant. No correlation was found with PSA, AFP, or CEA [50]. Ca-125 also alternates strongly with the diameter of IVC, as well as with the presence of fluid in the pleural cavity and peripheral edema [7]. This phenomenon was investigated by Yilmaz et al. in patients with end-stage kidney disease [1]. It correlated with the advancement of CKD, as well as with the levels of NT-pro-BNP and C-reactive protein and with a larger left ventricular end-diastolic diameter. The group of patients with a normal level of Ca-125 had higher albumin and hemoglobin levels compared to the group with elevated Ca-125. Serum NT-pro-BNP and Ca-125 were measured by a direct chemiluminescence assay. A correlation analysis between CA 125 levels and different parameters was performed, calculating Pearson’s or Spearman’s coefficient as appropriate. *p* < 0.05 was considered statistically significant. The authors excluded patients with pleural fluid accumulation which could influence the Ca-125 serum level. Núñez-Marín et al. found no correlation between IVC and Ca-125, but in their study, the carbohydrate antigen was independently associated with a congestive pattern of intrarenal venous flow [45]. This study also showed that not NT-pro-BNP but Ca-125 correlates with Doppler signs of volume overload, calculated using the Youden method. Carbohydrate antigen 125 appeared to increase with NT-pro-BNP, the ratio of 24 h peritoneal dialysate creatinine to serum creatinine, a decrease in albumin level, and the ECW/TBW ratio in an analysis including 489 adult patients on peritoneal dialysis [43]. The Ca-125 serum level was measured using an immune-assay sandwich assay. There was no correlation between the Ca-125 level and 24 h urinary creatinine clearance or CRP [43]. The researchers postulate that the observed decrease in albumin level was due to a dilution effect rather than a massive loss or cachexia. The marker has been found in peritoneal dialysate fluid and tends to increase during peritonitis not only in the PD population [4]. This is why authors do not think that Ca-125 should be a first-row OH marker for PD populations, as it tends to increase because of the presence of dialysate fluid in the abdomen, not due to total body water imbalance. Ca-125, despite being a cancer marker, has a good chance of becoming a fluid balance indicator for patients on HD or not treated with dialysis. The marker tends to increase particularly in right heart failure due to fluid accumulation, together with stimulation during intravascular congestion and consequent damage. During our literature research, we did not find specific data confirming that Ca-125 could identify the rapid and newly developed increase in fluid accumulation in the ESKD population. Overall, results are very promising; however, correlations are not always cohesive, and more studies are needed.

The used statistical techniques to compare Ca-125 level with other OH markers (together with *p* values) mentioned in the studies above are listed below in Table 4.

As can be clearly seen from Table 4, there are still not many studies that combine Ca-125 together with other OH markers. All the studies included NT-pro-NBP and one non-laboratory marker. In our opinion, further multi-way studies should be performed to develop the potential of Ca-125. The combination of Ca-125 together with BCM or USG could give better results in non-cancer patients than NT-pro-BNP, which is not a specific marker and tends to increase under variable conditions, which are not specifically related to OH. Natriuretic peptides increase in patients with cardiorenal syndrome even without fluid excess.

### 4.2. Available and Prospective OH Markers

As mentioned in Table 1, modern medicine can estimate volume status using a variety of the following methods.

### 4.3. Serum Markers

(a)NT-pro-BNP

N-terminal pro-B type natriuretic peptide (NT-pro-BNP) is a peptide hormone synthesized mainly by ventricular cardiomyocytes in response to stretching, e.g., during the increased cardiac filling pressure, and cleared by kidneys [51,52]. This molecule is produced particularly by stretched ventricular and atrial cardiomyocytes [33]. Elevated serum levels of NT-pro-BNP are observed in HF and during cardiac ischemia, pulmonary embolism, cor pulmonale, hypertension, hyperthyroidism, Cushing syndrome, hyperaldosteronism, cirrhosis, subarachnoid hemorrhage, and kidney failure. This marker also varies by sex and age and has lower values in obese patients. Its blood concentration can be affected by medications like corticosteroids, diuretics, ACE inhibitors, or thyroid hormones. Monitoring changes in NT-pro-BNP over time has been shown to be a strong diagnostic indicator, as life expectancy appears to increase as its concentration decreases [52]. NT-pro-BNP also increases and remains significantly higher in patients with accompanying ESKD [22,52]. In patients with CKD, HF is a dangerous clinical issue with insufficient treatment results. This is due not only to volume overload, but also to the development of anemia. All aforementioned pathologies cause an increase in both left ventricular end-diastolic volume and mass, which eventually leads to HF [53]. The African American Study of Kidney Disease and Hypertension enrolled patients with CKD to find an association between the risk of cardiovascular incidence and NT-pro-BNP levels in this population. It appeared that individuals with an increased plasma level of NT-pro-BNP were more likely to have cardiovascular complications, and this risk was particularly evident in patients with albuminuria [54]. Elevated NT-pro-BNP levels indicate an increased risk of cardiovascular events in the HD population with no other signs of HF according to the study by Goto et al. This marker was an independent risk factor, as it showed no correlation with age, body mass index, blood pressure, and heart rate [55]. Therefore, NT-pro-BNP, mainly used in cardiology, has attracted the attention of nephrologists. Its clinical application has also been discovered by nephrologists, as it tends to correlate with other OH indicators. In a meta-analysis involving 4287 patients, Schaub et al. asked whether NT-pro-BNP has a different diagnostic and prognostic utility in patients with kidney dysfunction. The correlation between GFR and natriuretic peptides was found to be statistically significant and ranged from −0.21 to −0.58, which means that during the decline in renal function, the NT-pro-BNP level increases [56]. An elevated serum level of this peptide in patients with kidney dysfunction compared to patients with normal NT-pro-BNP confers an increased risk of mortality when compared to healthy controls [56]. An independent relationship between eGFR and NT-pro-BNP was also observed in a study on 599 dyspneic patients with renal malfunction [57]. Analogous results were obtained by DeFilippi et al., and one-year mortality rates were 36.3% in patients with ESKD and 19.0% in patients without ESKD [58]. NT-pro-BNP showed a similar result in dialysis patients in a study by Park et al. [59]. Its level was significantly higher in patients with any type of dialysis treatment compared to the control group. NT-pro-BNP also correlated with AMD, even though BMC did not detect any differences in OH status between the control and treatment groups [59]. The marker increased together with the decrease in ejection fraction in HD patients who took part in Lee et al.’s study [47]. NT-pro-BNP increased significantly in patients who were labeled as overhydrated by BIS. Yilmaz et al.’s study on non-dialysis dependent patients indicated that log NT-pro-BNP increased together with OH/ECW calculated by BIS and remained significantly higher in patients who were defined as overhydrated [22]. This natriuretic peptide tends to increase in PD patients when compared to a healthy population, and scientists point out that the dialysis method has no influence on its level in the blood [59]. NT-pro-BNP tends to increase in individuals with fluid excess during clinical studies on various methods of estimating hydration status [22,30,32,42,43,44,49,60,61]. It was elevated during the increase in protein clearance during peritoneal dialysis [44], associated with pleural effusion and IVC diameter [30,61] and ECW/TBW ratio [32,42,43,60] but not with peripheral edema [19,30]. It has been compared in assessments of hydration status with Ca-125; in some studies, both markers were elevated with fluid excess [43], and in some cases, only Ca-125 increased [62]. Núñez-Marín et al. noticed that Ca-125 but not NT-pro-BNP correlated with VEXUS indicators of OH in patients with HF [45]. NT-pro-BNP correlated in establishing OH by BIS in Vega et al.’s study, along with a decrease in serum albumin, an increase in CRP, and proteinuria [25]. Fluid retention in patients with ESKD calculated by BIS corresponded to an increase in NT-pro-BNP, as serum levels in hypervolemia vs. euvolemia were 4.7 times higher [60]. In the study by Schork et al., NT-pro-BNP levels also corresponded to OH calculated using BIS in patients with ESKD [45]. In a research study involving 179 non-dialysis CKD patients in all stages, OH measured by BCM correlated with urinary protease activity and progression of renal dysfunction, as well as with increases in NT-pro-BNP and systolic blood pressure [42]. Schwermer et al. indicated that male gender, smoking, diabetes, and cardiovascular incidences were connected to OH [49]. Scientists noticed also that OH induced an increase in NT-pro-BNP and troponin concentration which was interpreted as cardiovascular toxicity of water surplus. In Lee et al.’s research, OH > 1 L as calculated by BIS correlated with an elevation of NT-pro-BNP in HD patients [47]. Similarly, in Drepper et al.’s study on PD patients, mortality was higher in those who were classified by BIS as overhydrated and with an increase in NT-pro-BNP serum level [23]. In general, clinicians must have the knowledge that there is a long list of factors that influence NT-pro-BNP concentration, including, for example, vitamin D status in HD patients [63].

(b)Adrenomedullin and proadrenomedullin

ADM is a peptide hormone synthesized by endothelial and vascular smooth muscle cells of organs like the lungs, brain, kidneys, heart, and adrenal medulla in response to an increase in fluid volume [64,65]. Its function is vasodilatation, preservation of endothelial integrity, and inhibition of the renin–angiotensin–aldosterone system (it protects the heart and kidneys from damage induced by angiotensin II) [64]. It tends to decrease during the use of diuretics, blockers of the RAA system, leading to the assumption that overhydration activates the sympathetic nervous system, which stimulates its production [64]. It has also been shown in experimental and epidemiological studies to have anti-inflammatory and antioxidant properties and the ability to reduce arterial intimal membrane hyperplasia when organs are exposed to damage [66]. ADM is significantly elevated in HF, sepsis, and other clinical states that lead to heart malfunction. The negative correlation between the elevation of ADM and a decrease in left ventricular ejection fraction was noted by Nishikimi et al. along with a positive correlation with an increase in NYHA class and NT-pro-BNP plasma level [67,68]. This marker can also be used by nephrologists to investigate its correlation with hydration status, not only in patients with concomitant HF, as it shows a very promising result in cardiological research. ADM is proportionally increased with the severity of kidney disease [67]. However, ADM is difficult to measure from a blood sample because it is rapidly removed from the circulation, and even when present in the bloodstream, it is covered by binding protein, making it inaccessible for immunometric analysis [50,64]. Pro-ADM is a precursor of ADM, the mid-regional fragment of which, called mid-regional ADM (MR-pro-AMD), is more stable and may directly reflect blood levels of adrenomedullin [66,67]. It seems to be a better predictor of 90-day mortality due to cardiac incidents than NT-pro-BNP, and its elevated level reflects poorer 12-month survival in patients with HF [67]. In Obineche et al.’s study, ADM remained high, together with NT-pro-BNP in PD patients [59].

It is also being studied in intensive care units among critically ill patients with septic shock and systemic inflammatory response syndrome [68]. MR-pro-AMD also correlated with the APACHE II score, SAPS II score, IL-6, creatinine, and age. In the ENVOL study, the proadrenomedullin indicator correlated strongly positively with sodium imbalance, OH, and current SOFA score [68]. In this study, only MR-pro-AMD and angiotensin II levels correlated significantly with sodium status, while pro-atrial natriuretic peptide (MR-pro-ANP), renin, aldosterone, cortisol, norepinephrine, epinephrine, copeptin, pro-endothelin, and EPO did not [69]. MR-pro-AMD was also studied in HD and PD populations for up to 7 years in Austria. The majority of patients (82%) included in the study had an elevated MR-pro-AMD level ≥ 1.895 nmol/L, and this was significantly higher in subjects who passed away during the study [70]. The peptide also correlated with another investigated marker, MR-pro-ANP, which was elevated in 99% of patients, and both parameters correlated with each other (r^2^ = 0.62). The two indicators were strongly related to the probability of death due to HF, but not within the entire group of fatal and non-fatal cardiovascular disease events. ADM seems to reflect the decompensated organ’s reaction to the multifractional injuries in preserving the integrity of the cardiovascular system in ESKD. MR-pro-AMD increased not only in patients with diagnosed HF, but also with the advancement of renal disease. MR-pro-AMD tended to correlate with a relative OH status in patients with both hemodialysis and peritoneal dialysis (n = 40) in Park et al.’s study. Its growth increased with the advancement of CKD, correlating significantly with NT-pro-BNP and cardiac markers (LV mass, LV mass index, ejection fraction, and left atrial diameter) [60]. These results give both ADM and MR-pro-ADM great potential to become independent indicators of OH.

(c)Galectin-3

The Gal-3 protein was discovered in the early 1980s, and since then, its role has been studied in several organs, including kidneys [71]. In pre-clinical models, it is overexpressed in diabetic nephropathy, toxic injury, cardiorenal syndrome, or ischemia/reperfusion injury. In renal carcinoma cells, Gal-3 shows that hypoxia is crucial for its expression, and its level elevates gradually with disease stage [72]. Gal-3 is also connected to immune-associated kidney damage like sepsis, cancer, or autoimmune diseases [71,73]. At the cellular level, Gal-3 is associated with renal fibrogenesis and chronic inflammation [74,75]. Pathomorphological analysis indicated that higher Gal-3 concentration is associated with interstitial fibrosis, tubular atrophy, and vascular intimal fibrosis. In a study on 198 patients who were treated with PD, an elevated level of Gal-3 corresponded to aortic stiffness, independently of age and gender [76]. In a 4-year clinical trial on 280 patients with renal disease, urinary Gal-3 also correlated negatively with eGFR and positively with proteinuria [77]. When considered as an OH marker, there is no direct connection, but the protein increases together with kidney and heart dysfunction due to overhydration. In an HF population, Gal-3 was associated with an increased risk of death after adjustment on a renal injury biomarker (*p* < 0.001) [78]. In an observational study of 1200 patients with HF, Gal-3 showed a negative correlation with eGFR, and a connection with a mortality risk when diminished renal function is present [79]. Patients with a higher Gal-3 concentration than the established mean value (23.2 ng/mL) had a higher mortality rate. However, it had no prognostic value as a mortality risk factor when renal function was preserved. It serves not only as a renal injury marker, but also as a heart injury indicator in the ESKD population. In a population of children on HD, Gal-3 increases along with left ventricular diastolic dysfunction [80]. The clinical guidelines announced by the American Heart Association/American College of Cardiology mentioned the utility of Gal-3 as a predictor of mortality and hospitalization in cases with HF [80]. This property makes Gal-3 a good marker to use both in ESKD when HF is suspected and vice versa. Zhang et al. found a correlation between Gal-3 and arterial wall stiffness in HD patients. Scientists discovered that Gal-3 increased in patients with log-transformed dialysis vintage, CKD progression, and mean arterial pressure [80]. Even if it is not a direct indicator of OH, it should be considered as a marker of renal disease due to its correlation with organ damage.

(d)Urocortin-2

Ucn-2 is a peptide that has a similar structure to the corticotropin-release factor and binds via its receptor CRHRH-2 [81]. This receptor is mainly found in the central nervous system, heart, and endothelial and smooth muscle cells of the systemic vasculature. Its actions on animal tissues include vasodilation, positive inotropic and chronotropic effects, and cardioprotective abilities [82]. An increase in Ucn-2 is seen in HF, left ventricular systolic dysfunction, non-ischemic dilated cardiomyopathy, and pulmonary arterial hypertension (PAH) [82]. The significant adverse effect is that it can cause a significant decrease in blood pressure, leading to worsening of renal function in patients with ESKD [82]. When its action was compared with metoprolol, it increased heart hemodynamic parameters due to its inotropic and chronotropic effects along with an increase in mean arterial pressure (MAP) [83]. This peptide’s activity on neurohormonal and renal function is still not well understood. Ucn-2 stimulates diuresis, increases creatinine clearance, and inhibits sodium retention, but this phenomenon, which is seen in animals, is not always present in humans [84]. Urocortin dilated renal arteries in rats, and the magnitude of this effect did not vary between animals’ genders, but it seems that the mechanism is different in females than in males [84]. Due to its potential to become a marker of HF, Ucn-2 is still undergoing tests on both models. A study on a group of eight healthy men confirmed the hemodynamic effect, as well as the ability to decrease MAP and vascular resistance and increase the left ventricular ejection fraction [85]. In a combined clinical and experimental study, Ucn-2 was able to decrease PAH, improve right ventricle function, and improve pulmonary circulation [86]. However, Ucn-2 plasma levels did not differ between the patients who suffered from PAH and the healthy group, but increased m-RNA expression was observed in people with right ventricle failure. It correlated negatively with IVC collapsibility [82]. When Ucn-2 is considered as an OH status marker or factor that can improve renal function, the results differ between studies. In a study of 12 sheep injected with mouse Ucn-2 (via a pulmonary artery catheter), there was a reduction in the effect of HF factors, as well as an improvement in renal function. It was able to decrease the MAP and left atrial pressure and suppress the production of cardiac remodeling factors (aldosterone, arginine vasopressin, and endothelin 1) [81]. A decrease in creatinine and sodium blood levels combined with an increase in urine output indicates an improvement in renal function. The same scientific group compared the effects of Ucn-2 on heart and kidney function in a different sheep model and in comparison with dobutamine [85]. Dobutamine and Ucn-2 improved renal function, but the significant sodium excretion was altered by Ucn-2. More interestingly, Ucn-2 decreased the overall OH status, while dobutamine increased it. It also gave better results in both HF and OH compared to the other drug. Similarl o what was found in the article mentioned above, in an animal study, Ucn-2 exhibited a better effect on diuresis, creatinine level, and sodium balance than furosemide. It was able to reduce renin, aldosterone, and vasopressin levels [87]. Heart function also improved. Ucn-2 attenuated furosemide function, which is a promising property as some patients with ESKD develop diuretic resistance. An experimental study on rats investigating the possible influence of Ucn-2 on renal dysfunction and injury caused by ischemia or reperfusion showed that it was unable to decrease organ failure [88]. Ucn-2 did not increase the creatinine clearance or stop anuria; a higher dose of this protein even caused a decrease in renal function. The opposite effect was observed in a human study by Chan et al., where Ucn-2 revised renal function and slashed RAA activity when compared to a placebo [89]. The treated group required a lower dose of furosemide, and the indirect OH marker NT-pro-BNP decreased after the infusion. Ucn-2 needs further study in the future, as it has shown good results in animal models.

### 4.4. Non-Laboratory Tests

(a)Gold standard: bioimpedance spectroscopy (BIS)

BIS usage in CKD

BIS, as a noninvasive tool, seems to be perfect for the estimation of OH status in patients as it provides information about body composition and water placement in the body. Even though it is quite expensive, requiring an additional cost of conservation and body stickers, it is commonly used in dialysis stations and intensive care units as it is a reliable method even if the patients cannot be weighed due to their severity of illness. CKD patients who do not need renal replacement therapy also can suffer from OH. Sun et al.’s research on 302 patients with CKD stages 1–4 showed that calculated OH correlated positively with LVH [90]. Vega et al.’s original paper showed that there is an association between OH status calculated by BIS and higher mortality in patients with ESKD stages 4–5 who are not yet undergoing kidney replacement therapy. Kaplan–Meier analysis confirmed higher mortality in patients with excessive overhydration [25]. However, even if it is easily available and used in renal diagnosis and dialysis maintenance, it does not reflect and correspond directly with hydration status [91,92]. Progression towards ESKD correlates with higher OH status. This interaction was observed by Hung et al. in a nearly three-year study of patients with CKD stages 3–5. OH appeared to be a more important mortality risk factor than hypertension. Even if assumed as a gold standard, it excludes patients with limb prostheses, heart stimulation, or metal joints.

BIS usage in dialysis patients

Volume status estimation is particularly significant for patients who undergo renal replacement therapy because it strongly correlates with patients’ wellbeing and mortality [93]. Oe et al.’s study on OH measurements by BIS on patients who were treated with both PD and HD revealed that patients are prone to OH independently of the dialysis method [94]. Another research paper, which included patients on HD, showed that OH calculated by BIS is an independent predictor of death in the dialysis population [14]. Unfortunately, patients with metallic joint prostheses, cardiac pacemakers, decompensated cirrhosis, and limb amputations were excluded from the study due to the limitations of the BIS technique. In Kim et al.’s study on a group of 147 HD patients, OH correlated negatively with creatinine, serum albumin, white blood cell count, platelets, uric acid, potassium, phosphorus, and triglycerides [95]. In a study by Siriopol et al. on HD patients [29], BIS did not improve life expectancy or help maintain dry weight. In a systematic review and meta-analysis by Covic et al. on 1312 ESKD patients, BIS-based dialysis therapy did not reduce all-time mortality. Together with this observation, BIS had no effect on body change, but improved systolic blood pressure [96]. In a study performed on peritoneal dialysis (PD) patients, BIS water balance calculations correlated with urine protein loss and higher creatinine, regardless of the duration of dialysis therapy [44]. Comparable dependency was noticed in a pediatric population independently of the dialysis method [32]. In a study on 13 children who underwent PD, the BIS calculations were superior to body weight measurements in assessing volume-dependent factors like blood pressure in patients with severe OH [97]. The authors point out that the best effects are obtained when ESKD patients have regular BIS measurements, not only when OH is suspected. In a meta-analysis by Wang et al. among almost 105,000 patients who underwent both types of renal replacement therapy, one-third had overhydration detected by BIS [12]. The estimated risk factor for mortality and cardiovascular events was ECW/TBW > 0.4. Scotland et al.’s randomized controlled trials on both HD and PD, which compared fluid management using BIS versus standard clinical methods (arterial stiffness, body weight, systolic blood pressure), estimated that spectroscopy better reflects hydration status [98]. In a review published in 2020, the authors point out that even though BIS calculations are based on various theoretical assumptions, the effectiveness of usage in HD patients remains quite promising [99]. However, the authors note that the noticeable benefits of BIS calculations in PD patients are limited, and further studies are needed.

In a prospective, observational study on a PD population, the association between OH and time of death or transfer to HD was clearly seen [100].

(b)Ultrasonography

Ultrasonography (US) is one of the common tests performed clinically either in a specific room or beside the patient’s bed—POCUS. In terms of hydration status, doctors can visualize and measure the width of IVC, jugular veins, hepatic portal vein, and renal veins. POCUS is nowadays one of the components of physical examination inextricably connected with auscultation, palpation, and inspection [101]. Other simple radiological examinations are inconclusive, as the presence of pleural fluid can only be observed by X-ray if at least 200 mL of fluid is present [17]. POCUS accelerates the diagnosis or exclusion of some pathologies in real time without the need for consultation [102]. The dependency between POCUS and hydration status can be divided according to the stages of renal disease, dialysis method, or HF.

Koratala et al. gave an example of a patient with CKD who had missed one dialysis session and suffered from shortness of breath; POCUS revealed fluid around his heart in the pericardium [102]. Lung POCUS can reveal extravascular fluid as a diffuse B-line pattern (otherwise known as a comet). Both symptomatic and asymptomatic forms of lung congestion worsen outcomes in patients with CKD [21]. Pulmonary congestion assessed by USG is noticeable in ESRD patients, independently of the dialysis method [103]. Lung USG is also used by nephrologists to guide dry body weight estimation during HD [103,104].

In Enia et al.’s study, almost 40% of HD patients with lung congestion were asymptomatic [105]. In the study guided by Lutradis et al., scientists studied the effect of the estimation of dry body weight based on lung USG for 8 weeks in ambulatory conditions [106]. It appeared that patients guided by this protocol maintained dry body weight better and exhibited a decrease in blood pressure compared to a group guided by normal criteria [106]. On the other hand, a study on 250 HD patients in whom dry body mass was adjusted using lung POCUS together with BIS did not improve life expectancy or cardiovascular events [29]. Extracellular fluid estimated by USG also correlates with the BIS method in HD patients [13]. Siriopol et al. concluded that both lung USG and BIS results are independently associated with all-cause mortality in an HD population, but only spectroscopy-assessed OH correlated with risk prediction for death beyond echocardiographic-based risk scores [46]. The presence of comets, which also indicates the presence of lung fluid, was significantly decreased after the hemodialysis session and was a better overhydration predictor than BIS itself [48]. Kaplan–Meier analysis showed that a higher mortality rate was present in patients with severe lung congestion. A limitation of lung POCUS in OH diagnosis is the fact that there is no one specific protocol available in the literature, and the fact that B-lines are not specific for pulmonary edema [101]. As noticed and described above, the width of IVC is not always an indicator of overhydration, as its dilatation has been found in both healthy athletes and those with diseases such as valvular and pulmonary hypertension [103]. IVC diameter is used to estimate the right atrial pressure, but it does not provide any data about the organ’s congestion [101]. The strong limitation of the lung USG is the fact that it reflects only left heart pressure but gives no information about venous congestion [101]. The measurements of hepatic vein flow via Doppler without a simultaneous performance of electrocardiography lead to several errors, as waveforms are influenced by heart arrhythmias [101]. The physician’s experience also plays a key role in assessing the hydration status of the patient, which may differ in patients with obesity or hyperventilation. Misinterpretation can also be caused by improper patient positioning or the presence of a catheter [30].

Table 5 includes different markers and whether they are sensitive, specific, or able to predict fluid imbalance. The potential cost of each test was estimated based on available prices for commercially available ELISA assays and calculated for 100 assays. Prices were checked on 31 December 2023.

When analyzing Table 5 we can notice that Ca-125 holds great promise for male patients as the results would not be influenced by a menstrual cycle or pregnancy. When this measurement is applied to PD patients, the abdomen should be empty, and inflammation must be excluded. Unfortunately, Ca-125 concentration is affected by the presence of dialysate fluid. NT-pro-BNP seems to be the cheapest blood test but has the least specificity for the OH status itself. Its serum level is influenced by both medications widely used by ESRD patients and isolated heart dysfunction. The marker cannot differentiate between HF arising from fluid excess and that from other causes like arrhythmia. The MR-pro-ADM serum level is affected by plenty of metabolic disturbances which are widely common in the population of the 21st century. However, it holds great promise because it seems to be less altered by HF or ongoing inflammation when compared to Gal-3, Ca-125, or NT-pro-BNP. Gal-3 seems to not be a perfect marker for patients after myocardial infarction, but similarly to Ucn-2, more studies are necessary to discover its practical use. The price of Ucn-2 is the highest per 100 studies, which could be because it is not a widely used marker in daily medicine. Other modalities are often hard to calculate, but they are limited to people without obesity or pregnancy. USG results are also influenced by the doctors’ skills and eventual training.

While considering the potential synergistic use of Ca-125 with other markers, it seems to correlate best with non-laboratory tests like BIS and USG. As BIS equipment is not common in every hospital ward, we recommend the combination of blood markers together with USG. The assessment of the IVC diameter together with laboratory test confirmation could give the best and most precise results in the estimation of OH. Before the blood intake, doctors should perform a short survey for the patients, to exclude eventual errors:-Is the patient pregnant?-Is the patient menstruating?-Is the patient suffering from HF?-Is the patient suffering from liver cirrhosis or cancer or undergoing chemotherapy?-Is the patient on HD or PD?

## 5. Conclusions

Ca-125 holds great promise to become a new OH marker as it is an easily available tool, and it does not require invasive intervention. The great potential of Ca-125 noticed by cardiologists should be investigated further by nephrologists not only in patients with HF.

The current understanding of Ca-125 as a marker triggered by the ongoing inflammation process and damage of mesothelial cells is directly related to OH as it triggers both mentioned factors.

Doctors must remember that this marker also increases in various clinical conditions that should be excluded like cancer, liver cirrhosis, or pelvic inflammation [108]. When diagnosed in women with ESKD, it may fluctuate during the menstrual cycle and can give imprecise results especially when assessed at the same time as menstruation [5]. More studies on multiple markers where Ca-125 will be evaluated together with other aforementioned substances are necessary.

Modern medicine has a huge need for OH markers and, as described in this review, there are no specific and accurate ones. The presence or absence of symptoms during a simple physical examination does not exclude overhydration. The combination of BIS and POCUS with the VEXUS protocol seems to have promising and good results. The addition of serum markers like NT-pro-BNP or Ca-125 broadens the viewpoint on volume status when compared to studies based on only one OH test. The markers mentioned in the study, namely ADM, pro-ADM, Ucn-2, and Gal-3, need to be further investigated as OH markers. Literature and clinic experience provide data indicating that patients’ lifestyle has a significant influence on OH. Low sodium intake by ESRD individuals diminishes kidney disease progression [33,104,109]. This phenomenon may be connected to angiotensin II and aldosterone effects. Recommendations about daily salt intake vary depending on race, age, and the presence of other illnesses, and clinicians should be aware of this fact [109]. Doctors must always remember the power of a healthy lifestyle and patients’ self-awareness when dealing with an OH issue.

## Data Availability

Not applicable.

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
