# Peer review of "On Whether Ca-125 Is the Answer for Diagnosing Overhydration, Particularly in End-Stage Kidney Disease Patients—A Systematic Review"

_ijms, 2024, doi:10.3390/ijms25042192_

Round 1

Reviewer 1 Report (Previous Reviewer 2)

Comments and Suggestions for Authors

Dear Authors,

Thank you for submitting your revised manuscript titled "Is Ca-125 the Answer for Overhydration Diagnosis Particularly in ESKD Patients – A Systematic Review" and for addressing the comments in a comprehensive manner.

The revisions made, particularly in the conclusion section, methodology clarification, and comparative analysis with other markers, have significantly improved the manuscript. Your effort in enhancing the statistical analysis details and addressing the potential biases is commendable. The additional context and implications provided in the discussion are beneficial for understanding the broader impact of your research.

Overall, the manuscript has made considerable progress. With attention to these final details, it will be a valuable contribution to the field.

Sincerely,

Author Response

Thank you for your letter and for the reviewers’ comments on our manuscript entitled “Is Ca-125 the answer for overhydration diagnosis particularly in ESKD patients – a systematic review”. All these comments were very helpful for revising and improving our paper. We have studied these comments carefully and have made corresponding corrections that we hope will meet with your approval. The changes in the revised manuscript are highlighted in blue. Responses to the reviewers’ comments are provided below.

We would like to express our great appreciation to you and the reviewers for the comments on our paper. If you have any further queries, please do not hesitate to contact us.

Kind regards,

Barbara Emilia Nikitiuk

Reviewer 2 Report (New Reviewer)

Comments and Suggestions for Authors

Dear authors you will find attached my comments.

Comments on the Quality of English Language

Moderate almost extensive  editing of English language required.

Author Response

C1 : Please make sure that permission has been obtained and there is no copyright issue (PRISMA flow diagram)

The prisma flow diagram had been labeled and the authors had been cited below. I will leave the information provided by the authors which were provided in the “PRISMA 2020 explanation and elaboration: Updated guidance and exemplars for reporting systematic reviews”

This is an Open Access article distributed in accordance with the terms of the Creative Commons Attribution (CC BY 4.0) license, which permits others to distribute, remix, adapt and build upon this work, for commercial use, provided the original work is properly cited. See: http://creativecommons.org/licenses/by/4.0/

C2 : Regarding references, I think you should also add “pediatric”  references, e.g

Karava V, Stabouli S, Dotis J, Liakopoulos V, Papachristou F, Printza N. Tracking hydration status changes by bioimpedance spectroscopy in children on peritoneal dialysis. Peritoneal Dialysis International. 2021;41(2):217-225. doi:10.1177/0896860820945813

Thank you very much for this extremley valuable research. We included it in our systematic review.

C3 : Line 54 please rephrase

Line 54 had been rephrased.

C4 :..” In a Eng et al. study on pediatric ESKD population, 25% of children were hypertensive but not overhydrated, which proves that simple blood pressure measurement is not enough to  estimate the hydration status” It is common knowledge that  hypertension is not the result of  just overhydration  but result  of multiple factors and its presence or absence is not linearly related to hydration status. Please rephrase or explain.

The piece of text cited above had been rephrased and explained.

C5: Υou mention that the technical equipment for BIS is expensive but it should be mentioned that it is an essential tool for nephrology units and intensive care units as it is a reliable method even in patients who cannot be weighed.

The information about the BIS usage in nephrology as well as intensive care units had been included in lines 499-502.

C6:  CA-125 is known to be a marker of peritoneal sclerosis in ESKD patients on PD  (  results to inadequate clearance). Therefore, in this category of patients it would probably not be a useful indicator of overhydration.

Thank you very much for this comment. Since Ca-125 production is stimulated by the presence of peritoneal fluid during PD, patients should be excluded from the study. This fact had been described in lines 246-249 and 646.

Reviewer 3 Report (New Reviewer)

Comments and Suggestions for Authors

First of all, we would like to thank the authors for their updated review of the different markers of overhydration in ESKD. This is an extremely relevant issue today, where there is no precise marker to assess overhydration. They mention the most relevant studies on the use of BIS, US, NT-pro-BNP and the new markers such as Ca-125, Gal-3, Adrenomedullin and UCN-2.

The review is generally well written and contains all the main information on the topic.

However, I would like to make some recommendations to improve the quality of the manuscript.

Unfortunately it brings together many instrumental methods and clinical biomarkers to assess intravascular and systemic congestion, making the text very long and tedious to read. Furthermore, the title is focused on ESKD, and during the review there is mention of both early stage CKD, ESKD and KRT (HD, PD). I feel it is too long. I would recommend mainly reducing paragraphs that do not provide relevant data such as those that mention early stage CKD. And I also think that the authors are very enthusiastic about the use of Ca-125, which currently recent works speak more of a marker of intravascular congestion by right CHF and long-term, not as an acute marker. Please comment on this.

Comments on the Quality of English Language

Moderate editing of English language required

Author Response

Thank you very much for all of the feedback provided. We will adress to all of your comments. 

The paragraphs about BIS, ADM/pro-ADM, Ucn-2 had been shortened. We excluded studies about the early stages of CKD to a minimum. The comment about the lack of evidence which will allow medics to use Ca-125 as an acute OH marker is in the line 250-254.

English had been revised once again and improved.

Round 2

Reviewer 2 Report (New Reviewer)

Comments and Suggestions for Authors

Dear authors , your revised manuscript  , in my opinion , can be accepted in this form. 

Comments on the Quality of English Language

Minor editing is needed. 

Reviewer 3 Report (New Reviewer)

Comments and Suggestions for Authors

The authors have responded correctly to the recommended suggestions. In my opinion, the manuscript is much improved, and could be published in this format. 

This manuscript is a resubmission of an earlier submission. The following is a list of the peer review reports and author responses from that submission.

Round 1

Reviewer 1 Report

Comments and Suggestions for Authors

This work brings together all the methods to evaluate the state of hydration in the subject on chronic dialysis. Discussing together instrumental methods and biomarkers does not seem very useful, lengthens the text and does not bring clarity. I would suggest that you rewrite the text focusing only on possible biomarkers, discussing in detail the merits and defects, as well as the costs for a possible frequent use. Despite the possible implications of the use of the CA-125, in general the Authors seem too enthusiastic about the considerable limitations still in place. 

Comments on the Quality of English Language

 Need improvement

Reviewer 2 Report

Comments and Suggestions for Authors

Dear Authors,

I appreciate the opportunity to review your manuscript titled "Is Ca-125 the Answer for Overhydration Diagnosis Particularly in ESKD Patients – A Systematic Review" for the Journal of Clinical Medicine. Your work addresses a critical and under-explored area in nephrology. Below, I elaborate on the areas for improvement to further enhance the impact and clarity of your study.

Strengths:

  1. Innovative Topic: Your manuscript addresses a novel and clinically significant question about the use of Ca-125 in diagnosing overhydration in ESKD patients. This innovative angle provides valuable insights into nephrology diagnostics.

  2. Depth of Research: The extensive review of literature across multiple disciplines is commendable. Your work synthesizes a wide range of studies, offering a comprehensive overview of the topic.

  3. Clinical Relevance: The subject of your research holds substantial clinical importance. By exploring the potential of Ca-125 as a diagnostic tool, your study could influence future medical practices in the management of ESKD.

  4. Interdisciplinary Approach: Your manuscript adeptly integrates knowledge from nephrology, cardiology, and oncology, enriching the understanding of overhydration in kidney diseases.

Expanded Areas for Improvement:

  1. Cohesive and Integrative Conclusions:

    • The current conclusion section presents findings in a somewhat disjointed manner. Integrating these into a more cohesive narrative would significantly enhance the paper's impact. Specifically, a clearer articulation of how the findings contribute to the current understanding of Ca-125's role in overhydration diagnosis in ESKD patients is needed.
    • Consider providing a summarizing table or model that integrates the diverse findings from the literature into a unified framework.
  2. Methodological Clarification and Rigor:

    • The criteria for study selection and exclusion in the systematic review require further detail. This includes more information on the databases searched, keywords used, and the time frame of the literature search.
    • The process of data extraction and synthesis could be more transparent. Clarifying how the data from different studies were compared and analyzed would strengthen the methodological rigor.
    • Address potential biases in the selection of studies and discuss the generalizability of the findings.
  3. Comparative Analysis with Other Markers:

    • A more comprehensive comparison with other overhydration markers would provide context and demonstrate the relative effectiveness of Ca-125. This could involve a side-by-side analysis of sensitivity, specificity, and predictive values.
    • Investigate the potential synergistic use of Ca-125 with other markers. Could a combination of markers provide a more accurate diagnosis of overhydration in ESKD patients?
  4. Statistical Analysis and Interpretation:

    • Some of the statistical analyses and their interpretations in the paper could be more robust. Detailing the statistical tests used and justifying their choice would enhance the paper's scientific rigor.
    • Discuss the limitations of the statistical approaches taken and how they might affect the interpretation of the results.
  5. Broader Context and Implications:

    • The discussion could benefit from a broader view of the implications of your findings in clinical practice. How might this research change current diagnostic or treatment protocols in ESKD?
    • Address the practical aspects of implementing Ca-125 as a diagnostic tool in clinical settings. Consider cost, accessibility, and the potential need for new protocols or training.

English Language and Formatting:

While your manuscript is generally well-structured and informative, there are several areas where language and formatting could be improved for clarity and professional presentation:

  1. Grammatical and Syntax Errors:

    • Instances of awkward phrasing and sentence construction are present, which could hinder the reader's comprehension. For example, in the abstract, phrases like "Patients who required kidney replacement therapy suffer from water imbalance..." could be rephrased for clarity.
    • There are occasional grammatical errors, such as improper tense usage and subject-verb agreement. These should be corrected to maintain academic standards.
  2. Consistency in Terminology:

    • Ensure consistency in the use of medical terms throughout the manuscript. For instance, the terms 'overhydration' and 'fluid overload' are used interchangeably; a consistent terminology would enhance clarity.
  3. Formatting Issues:

    • There are some inconsistencies in formatting, especially in the references and tables. Uniform formatting according to the journal's guidelines will improve the professional presentation of your manuscript.
    • Some tables and figures are not clearly labeled or referenced in the text, which could confuse readers.

I recommend a detailed proofreading by a native English speaker to ensure clarity and fluency in the language used. This includes checking for grammatical errors, improving sentence structure, and ensuring consistency in terminology.

By addressing these recommendations, particularly the linguistic and formatting aspects, your manuscript's quality and readability will significantly improve. I believe that your research is of great value to the field and look forward to seeing its enhanced version.

Sincerely,

Comments on the Quality of English Language

English Language and Formatting:

While your manuscript is generally well-structured and informative, there are several areas where language and formatting could be improved for clarity and professional presentation:

  1. Grammatical and Syntax Errors:

    • Instances of awkward phrasing and sentence construction are present, which could hinder the reader's comprehension. For example, in the abstract, phrases like "Patients who required kidney replacement therapy suffer from water imbalance..." could be rephrased for clarity.
    • There are occasional grammatical errors, such as improper tense usage and subject-verb agreement. These should be corrected to maintain academic standards.
  2. Consistency in Terminology:

    • Ensure consistency in the use of medical terms throughout the manuscript. For instance, the terms 'overhydration' and 'fluid overload' are used interchangeably; a consistent terminology would enhance clarity.
  3. Formatting Issues:

    • There are some inconsistencies in formatting, especially in the references and tables. Uniform formatting according to the journal's guidelines will improve the professional presentation of your manuscript.
    • Some tables and figures are not clearly labeled or referenced in the text, which could confuse readers.

We recommend a detailed proofreading by a native English speaker to ensure clarity and fluency in the language used. This includes checking for grammatical errors, improving sentence structure, and ensuring consistency in terminology.